# The Function and Therapeutic Potential of lncRNAs in Cardiac Fibrosis

**DOI:** 10.3390/biology12020154

**Published:** 2023-01-19

**Authors:** Xiang Nie, Jiahui Fan, Dao Wen Wang

**Affiliations:** 1Division of Cardiology, Department of Internal Medicine, Tongji Hospital, Tongji Medical College, Huazhong University of Science and Technology, Wuhan 430030, China; niexiang@tjh.tjmu.edu.cn (X.N.); fanjiahui@tjh.tjmu.edu.cn (J.F.); 2Hubei Key Laboratory of Genetics and Molecular Mechanisms of Cardiological Disorders, Huazhong University of Science and Technology, Wuhan 430030, China

**Keywords:** cardiac fibrosis, lncRNAs, TGF-β, ECMs, exosome

## Abstract

**Simple Summary:**

Cardiac fibrosis is a vital pathophysiologic change in heart disease, which eventually leads to heart failure. Several molecular mechanisms positively or negatively regulate myocardial fibrosis, among which long noncoding RNAs have gained increased attention. We summarize the contributions of lncRNAs to miRNA expression, TGF-β signaling, and ECMs synthesis, with a particular attention on the exosome-derived lncRNAs in the regulation of adverse fibrosis as well as the mode of action of lncRNAs secreted into exosomes. We also discuss how the current knowledge on lncRNAs can be applied to develop novel therapeutic strategies. This study may provide clues for the prevention and therapy of cardiac fibrosis.

**Abstract:**

Cardiac fibrosis remains an unresolved problem in cardiovascular diseases. Fibrosis of the myocardium plays a key role in the clinical outcomes of patients with heart injuries. Moderate fibrosis is favorable for cardiac structure maintaining and contractile force transmission, whereas adverse fibrosis generally progresses to ventricular remodeling and cardiac systolic or diastolic dysfunction. The molecular mechanisms involved in these processes are multifactorial and complex. Several molecular mechanisms, such as TGF-β signaling pathway, extracellular matrix (ECM) synthesis and degradation, and non-coding RNAs, positively or negatively regulate myocardial fibrosis. Long noncoding RNAs (lncRNAs) have emerged as significant mediators in gene regulation in cardiovascular diseases. Recent studies have demonstrated that lncRNAs are crucial in genetic programming and gene expression during myocardial fibrosis. We summarize the function of lncRNAs in cardiac fibrosis and their contributions to miRNA expression, TGF-β signaling, and ECMs synthesis, with a particular attention on the exosome-derived lncRNAs in the regulation of adverse fibrosis as well as the mode of action of lncRNAs secreted into exosomes. We also discuss how the current knowledge on lncRNAs can be applied to develop novel therapeutic strategies to prevent or reverse cardiac fibrosis.

## 1. Introduction

The essential function of the heart is to supply sufficient blood to peripheral organs and tissues during both normal and stress conditions. Normal arrangement of individual cardiomyocytes and constant transmission of contractile force are necessary to maintain cardiac structure and function. Cardiac fibrosis is initially beneficial for cardiac structure maintaining and contractility through controlling the arrangement of cardiomyocytes and maintaining normal structure of left ventricle. However, sustained fibrosis will lead to stiffness of the ventricular wall and generally progress to deterioration of both cardiac systolic and diastolic function [1,2]. The pathophysiologic mechanisms of cardiac fibrosis are diverse and complex [3]. In myocardial infarction or myocarditis, necrotic cardiomyocytes are usually replaced by scar tissue. This type of fibrosis is considered as replacement fibrosis, which reflects the regenerative capacity of the heart upon injury. In addition to cardiac structural damage, replacement fibrosis predominantly reduces systolic dysfunction as compared with diastolic function [4]. The other type of cardiac fibrosis is diffuse myocardial fibrosis, which is characterized by the excessive deposition of collagen fibers (such as collagen I, II, III, and IV) or other extracellular matrix proteins such as fibronectin (FN) and matrix metalloproteinases (MMPs). This type of fibrosis is usually present in chronic cardiovascular diseases, such as hypertension, diabetic cardiomyopathy, atrial fibrillation, and hypertrophic cardiomyopathy [1,5,6,7]. On the converse of replacement fibrosis, diffuse myocardial fibrosis typically reduces left ventricular diastolic function and is less pronounced on systolic function [8]. The differential pathophysiology and clinical course of cardiac fibrosis resulting from the changed structural quality and various fibrillary composition should be considered separately when exploring their pathophysiological processes [9,10].

Activated cardiac fibroblasts (CFs) as well as myofibroblasts are the main cellular effectors in the occurrence and progression of cardiac fibrosis. CFs are a subclass of interstitial cells, which produce several ECM proteins, including collagens of types I, III, IV, and V [11,12]. Under normal conditions, the ECMs not only serve as a mechanical scaffold to maintain cardiomyocyte architecture, but also control the transmission of contractile force [13]. In the injured heart, the proliferation and differentiation of fibroblasts into myofibroblasts lead to excessive ECMs synthesis and deposition, which disrupt the arrangement of myocytes and conversely attenuate the transmission of contractile force [14,15]. In addition, the increased proliferation of fibroblasts or myofibroblasts also contribute to the progress of fibrosis. Apart from the fibroblasts, other cell populations, such as myocytes, endothelial cells, and inflammatory cells, etc., may also contribute to the pathogenesis of myocardial fibrosis [16,17]. Fibroblasts dynamically interact with the various cardiac cell populations in various ways, such as mechanical, chemical, and electrophysiological, to alter gene expression and cellular behavior [18]. Endothelial cells, immune cells, and vascular smooth muscle cells also secrete molecular signals that regulate the growth or apoptosis of CFs and control fibroblast behaviors or gene expression [16]. Myocardial fibrosis is multifactorial, with various cell populations, factors, and signaling pathways involved.

Over the past few years, various noncoding RNAs, such as miRNAs, circRNAs, and lncRNAs, have been identified with significant roles in cardiac fibrosis. Long non-coding RNAs (lncRNAs) are a class of nucleic acids with lengths of more than 200 nt which have no capacity to code proteins or peptides. They play crucial roles in the progress of embryonic development, cell growth, and differentiation through regulating gene expression at the transcriptional level, post-transcriptional level, or maintaining the stability of RNAs or proteins [19,20]. To date, with development of next-generation sequencing, an increasing number of lncRNAs have been identified with significant roles in the pathogenesis of myocardial fibrosis [21,22,23]. In this study, we summarize the functions and mechanisms of lncRNAs in cardiac fibrosis and discuss the underlying signaling pathways, with a particular emphasis on the exosome-derived lncRNAs in regulation of fibrosis as well as the mode of lncRNAs secreted into exosomes. Better understanding of the functions of lncRNAs might lead to novel therapeutic approaches for reversing cardiac fibrosis and preventing detrimental outcomes.

## 2. Characteristics of lncRNAs

The ENCODE project indicates that up to 80% of the human genome do not have the function of coding proteins, which may play a significant role in regulating gene expression [24]. The transcripts of most of these genes are non-coding RNAs (ncRNAs) including rRNAs, tRNAs, circRNAs, microRNAs, and lncRNAs, etc. [25,26,27]. The FANTOM consortium produces a comprehensive picture of the mammalian transcriptome and publishes 34,030 lncRNAs in mice based on the cDNA sequencing [28]. The number of known human lncRNA transcripts are over 173,000 based on data from the NONCODE database [29,30]. According to the association with annotated protein-coding genes, lncRNAs are classified as intergenic transcripts, and sense or antisense transcripts that overlap other coding genes [31,32]. According to the association with subcellular structures, lncRNAs are also classified as chromatin-associated RNAs, chromatin-interlinking RNAs, nuclear bodies associated RNAs and cytoplasmic transcripts [32,33,34] (Figure 1). Although there are a high number of lncRNAs continually being identified, most are not validated and their functions are largely unknown.

LncRNAs have a great diversity of important functions in body development, regulation of the cell cycle and apoptosis, cellular metabolism, inflammatory response, and tissue homeostasis [35,36,37]. They have been found with vital roles in mammalian gene regulation through various pathways, such as genomic imprinting, chromatin modification, mRNA decay, protein translation, and miRNA sponges [38,39,40,41]. Nuclear lncRNAs usually regulate gene expression through chromatin modification. For instance, the X inactive-specific transcript (Xist) gene regulates mammal X chromosome activation through producing a long noncoding RNA that modifies underlying chromatin and reduces X-linked gene expression [42,43,44]. LncRNA NORAD controls genomic stability through sequestering PUMILIO (pumilio-fem3-binding factor) proteins, which represses the stability and translation of mRNAs. Silencing of NORAD, PUMILIO drives chromosomal instability by inhibiting DNA repair [45,46]. Some lncRNAs also control gene expression by regulating the stability of mRNAs or modification of proteins. For example, lncRNA ZFAS1 is elevated in colorectal cancer. Knockdown of ZFAS1 decreases the RNA stabilization of SNORD12C/78 and NOP58 through binding with snoRNP to induce 2′-O-Me of 28S rRNA, which eventually inhibits the proliferation and invasion of colorectal cancer cells [47]. LncRNA LINRIS is up-regulated in colorectal cancers. LINRIS reduces IGF2BP2 mRNA expression levels by inhibiting the ubiquitin of IGF2BP2 on K139 sites to maintain its stability [48]. Another function of lncRNA is the competitive endogenous RNA (ceRNA). LncRNAs might suppress the activity of microRNA (miRNA) through serving as sponge RNAs, which lead to miRNA target gene expression increasing [49]. A new lncRNA, named MAR1 (muscle anabolic regulator 1), is significantly up-regulated in myogenesis. MAR1 enhances skeletal muscle strength by sponging miR-487b to regulate Wnt5A expression [50]. Another example of this type of lncRNA is Mirf. Silencing of Mirf promotes autophagy by reducing miR-26a expression in vivo and in vitro [51]. Although non-protein coding potential is a significant characteristic of lncRNAs, some of them might function in biological processes through producing peptides or proteins [52]. LncRNA DWORF (dwarf open reading frame) encodes a peptide of 34 amino acids. Silencing of DWORF in skeletal muscle inhibits Ca^2+^ clearance and suppresses the Ca^2+^ adenosine triphosphatase activity [53]. 

In the cardiovascular system, several lncRNAs, such as H19, HOTAIR, MIAT, etc., are abundantly expressed in myocardial tissues [37,54,55,56]. Previous studies have detected and characterized the expression and function of lncRNAs under physiological conditions or in disease states [57,58,59]. Several lncRNAs have been found with potential roles in heart disease or their expression levels are correlated to disease progression, especially in cardiac fibrosis [60,61]. In this review, the functions, mechanisms, and therapeutic potential of lncRNAs in regulating myocardial fibrosis are summarized and discussed in detail.

### 2.1. LncRNAs Serve as ceRNAs in Controlling Cardiac Fibrosis 

Competing endogenous RNAs (ceRNAs) usually regulate gene expression via sponging microRNAs (miRNAs) at the post-transcriptional levels. LncRNAs might serve as ceRNAs to control miRNA expression and subsequently regulate mRNAs translation and degradation [62]. Homeostatic imbalance in lncRNA–miRNA interaction results in physiological alterations inside the cells and tissues leading to the occurrence of the heart disease [23,63]. To date, significant functions of lncRNA–miRNA interaction in the pathogenesis of myocardial fibrosis are reported by several studies (Figure 2).

In diabetic cardiomyopathy (DCM) [64], lncRNA Kcnq1ot1 (KCNQ1 antisense transcript 1) is significantly up-regulated in myocardial tissues or cardiac fibroblasts treated with glucose. Silencing of Kcnq1ot1 represses the TGF-β signaling pathway via sponging miR-214-3p to suppress the target gene caspase-1 expression [65]. Another example of lncRNA–miRNA interaction in DCM is lncRNA GAS5. Knockdown of GAS5 efficiently attenuates cardiomyocyte injury and myocardial fibrosis via negatively regulating miR-26a/b-5p expression [66]. Similar molecular mechanisms are also observed in myocardial infarction (MI) [67]. Pro-fibrotic lncRNA (PFL) expression is elevated in the heart of mice induced by MI. Overexpression of PFL promotes cardiac fibrosis through increasing the viability of CFs and promoting the transition of fibroblast into myofibroblast via sponging let-7d, leading to increased platelet-activating factor receptor (PTAFR) expression [68]. LncRNA small nuclear RNA host gene 7 (SNHG7) is up-regulated in the infarcted area from left ventricle of mice after MI. Luciferase assay indicates that SNHG7 function through sponging miR-34-5p, which leads to the increased expression of ROCK1 (Rho-associated, coiled-coil domain containing protein kinases) [69]. LncRNA–miRNA interaction is also reported in atrial fibrillation (AF). Increased expression of lncRNA plasmacytoma variant translocation 1 (PVT1) is detected in AF and positively correlated with collagens expression levels. PVT1 overexpression aggravates Ang-II-induced atrial fibroblasts proliferation, collagens production, and TGF-β1 signaling activation by sponging miR-128-3p to facilitate specificity protein 1 (Sp1) expression [70,71]. LncRNA nuclear-enriched abundant transcript 1 (NEAT1) is up-regulated in atrial tissues of AF patients. NEAT1 knockdown improves Ang II-induced mouse atrial fibrosis via negatively regulating miR-320 expression leading to the up-regulation of neuronal PAS domain protein 2 (NPAS2) [72]. The above studies have revealed specific lncRNAs which when elevated function as miRNA sponges to mediate fibrotic processes associated with varied cardiac diseases.

However, the underlying mechanisms are largely undefined, and some concerns should be under consideration when exploring the role of lncRNA–miRNA interaction in myocardial fibrosis. Firstly, the physiological interactions between lncRNAs and miRNAs in normal CFs are rarely illuminated. LncRNAs, down-regulated in myocardial fibrosis, might disturb the physiological lncRNA–miRNA interaction networks, thus promoting the expression of miRNAs, and subsequently repressing the downstream mRNA translation. Secondly, the abundance of lncRNAs resident in the cells should be sufficient. Low levels of lncRNAs might fail to completely sponge high abundant miRNAs. Moreover, ceRNA activity is also influenced by other multiple factors such as the subcellular localization of ceRNA components, binding affinity of miRNAs to lncRNAs, RNA secondary structures, and RNA-binding proteins [73,74,75].

### 2.2. LncRNAs Regulate Cardiac Fibrosis through TGF-β Signaling Pathways

Transforming growth factor β (TGF-β) stimulation triggers CFs proliferation and activation, including ECM proteins synthesis and deposition as well as fibroblast-to-myofibroblast differentiation [76,77]. The TGF-β receptor is a dimeric receptor complex whose activation promotes the phosphorylation of Smad2/3 transcription factors through the canonical signaling pathway [78]. Phosphorylated Smads (Smad2, 3, and 4) transfer signal messages to the nucleus and promote gene transcription [79,80]. Fibroblast-specific silencing of TGF-β receptors markedly reduce the pressure overload-induced fibrotic response [81,82,83]. Knockout of Smad2/3 in fibroblasts reduces the expression of fibrosis-related genes and alleviates injury-induced cellular proliferation within the heart [84,85].

The essential interplay between lncRNA and TGF-β signaling has been widely reported [86]. TGF-β upregulates lncRNA expression in various cancers such as lung cancer, breast cancer, and hepatocellular carcinoma [87,88,89]. In addition to being effectors of TGF-β signaling, several lncRNAs are reported to regulate TGF-β signaling pathway through various mechanisms. For example, lncRNA LINC00941 stimulates epithelial-mesenchymal transition by directly binding with Smad4 and competing with β-trcp (beta-transducing repeat containing E3 ubiquitin) to prevent the degradation of Smad4 protein, which eventually activates the TGF-β signaling pathway [90]. However, the interaction between lncRNAs and TGF-β signaling in myocardial fibrosis is rarely discussed in detail.

In myocardial fibrosis, TGF-β stimulates several lncRNAs expression in vitro and in vivo. The increased expression of Neat1 is detected in the heart tissue from transverse aortic constriction surgery-induced mice and TGF-β1 treated cardiac fibroblasts. Neat1 recruits Ezh2 to the promoter of Smad7 resulting in decreased Smad7 expression [91]. LncRNA Safe is up-regulated in TGF-β-induced cardiac fibrosis and myocardial infarction [92]. However, the underlying mechanism of TGF-β in regulating lncRNAs expression is not currently known. Smad2/3 proteins phosphorylated by TGF-β might be involved in provoking the transcription of lncRNAs. In addition, activation of the TGF-β receptors also initiate noncanonical signaling to promote the activation of the MAPK, p38, JNK1/2, and ERK1/2 signaling pathways [93,94], which are crucial factors in regulating fibrosis-associated gene expression. More work is still required to answer these questions.

Conversely, lncRNAs also stimulate the activation of TGF-β signaling pathway. The expression of long noncoding RNA AK081284 is up-regulated in cardiac fibroblasts treated with IL-17 or high glucose. Overexpression of AK081284 in cardiac fibroblasts promotes the production of collagens and TGF-β1, while AK081284 silencing reduces collagen and TGF-β1 expression [95]. LncRNA Cfast (cardiac fibroblast-associated transcript) is significantly up-regulated during myocardial infarction. Silencing of Cfast results in reduction of fibrosis-related gene expression and the transdifferentiation of myofibroblasts into fibroblasts. Cfast inhibits the interaction between COTL1 (coactosin-Like protein 1) and TRAP1 (transforming growth factor-β receptor-associated protein 1), which eventually activates the TGF-β signaling pathway [96]. In addition, overexpression of lncRNA GAS5 suppresses TGF-β-induced fibroblast to myofibroblast differentiation. GAS5 directly binds and promotes SMAD3 binding to protein phosphatase 1A (PPM1A), and thus accelerates SMAD3 dephosphorylation in fibroblasts induced by TGF-β [97]. Increased expression of lncRNA Safe is detected in fibrotic ventricular tissues induced by myocardial infarction. Knockdown of Safe prevents TGF-β-induced fibroblast to myofibroblast transition and extracellular matrix proteins production by inhibiting neighboring gene SFRP2 (secreted frizzled-related protein 2) expression [92]. 

The potential mechanisms involved in these processes must also be systematically explored. LncRNAs resident in the nuclear of fibroblasts may regulate TGF-β signaling pathway-associated gene expression through epigenic or transcriptional regulation. Cytoplasmic lncRNAs may regulate gene expression by controlling the translation or metabolism of RNAs or proteins.

### 2.3. LncRNAs Control Cardiac Fibrosis by Regulating ECM Gene Expression

In myocardium, excessive synthesis and deposition of extracellular matrix (ECM) proteins are the significant characteristics of myocardial fibrosis. Cardiac ECMs are primarily composed of fibrillar collagens, especially for type I and III, which are the principal proteins in maintaining cardiac structure and function [98]. Cardiac ECMs also contain nonstructural matricellular glycoproteins, proteoglycans, and glycosaminoglycans. The synthesis and degradation of ECMs are predominately regulated by metalloproteinases (MMPs) [99]. The balance between ECMs synthesis and degradation is of crucial importance in cardiac structural integrity and formation of fibrosis. However, to date, few studies systematically elaborate the functions and mechanisms of lncRNAs in these processes.

In myocardial fibrosis, lncRNAs display potential functions in regulating the expression of ECM genes. Bioinformatics analysis indicates that the differentially expressed lncRNAs and extracellular matrix (ECM) protein coding genes revealed a strong association between lncRNAs and ECMs [100]. LncRNA H19 directly binds and antagonizes YB-1 (Y-Box binding protein 1) under hypoxia, which results in the de-repression of collagen 1A expression and cardiac fibrosis [101]. Myocardial infarction associated transcript (MIAT) is up-regulated in myocardial infarction heart tissues. Down-regulation of MIAT alleviates cardiac fibrosis and improves cardiac function by regulating the expression of the fibrosis-related regulators [102]. LncRNA Wisper (Wisp2 super-enhancer-associated RNA) expression is enriched in CFs and elevated in a murine model of MI. Wisper regulates cardiac fibroblasts survival and behavior by regulating lysyl hydroxylase 2 expression [70]. Silencing of lncRNA Meg3 prevents cardiac MMP-2 production, decreases cardiac fibrosis, and improves diastolic function in mice induced by transverse aortic constriction surgery [103]. Knockdown of lncRNA MALAT1 prevents fibroblast proliferation, and ECMs production in AngII-treated cardiac fibroblasts [23].

Although these studies indicate that ECM genes are dysregulated upon lncRNAs stimulation, it seems more likely that the synthesis of ECMs is a common change of myocardial fibrosis induced by lncRNAs. Whether lncRNA have a direct role in controlling ECM genes expression is still a mystery. 

### 2.4. Exosome-Derived LncRNAs Regulate Cardiac Fibrosis

A number of studies have indicated that lncRNAs significantly regulate fibrosis by being expressed within fibroblasts and have a direct effect on ECM gene expression, TGF-β signaling pathway, and proliferation of fibroblasts or transition to myofibroblast. These effects are also proposed to be mediated by paracrine communication between donor and recipient cells, especially in cardiomyocytes and fibroblasts. Increasing evidence attaches much importance of noncoding RNAs, such as miRNAs, circRNAs, and lncRNAs, in the communication between cells by way of extracellular vesicle-mediated transfer from donor cells to the recipient cells [104,105,106,107]. This has been considered an important behavior of cardiac cells to communicate with each other and respond to cardiac injuries [108,109]. Previous studies have already confirmed the existence of exosomes in heart tissues and vessel walls using electron micrographs [110]. Fibroblasts and cardiomyocytes might interact with each other through the transfer of extracellular vesicles containing lncRNAs [111]. 

Involvement of lncRNAs in the crosstalk between cardiac fibroblasts and other cell populations have already been reported in recent years. Exosomes-containing lncRNA ZFAS1 induces cardiac fibrosis via the Wnt4/β-catenin signal pathway by sponging miR-4711-5p in cardiac fibroblasts [112]. LncRNA MIAT is up-regulated in serum-derived extracellular vesicles (EVs) from AF patients. MIAT aggravates the atrial remodeling and promotes AF by binding with miR-485-5p [113]. Neat1 is obviously up-regulated by P53 and HIF2A in cardiomyocytes in response to hypoxia and is enriched in cardiomyocyte-derived exosomes. Neat1 is essential for cell survival and fibroblast functions. Genetic knockout of Neat1 impairs cardiac function during myocardial infarction [111] (Figure 3 and Table 1).

### 2.5. The Way of LncRNAs Secreted into Exosomes

A variety of vesicles with different sizes and contents have been identified in the eukaryotic cells and tissues, including exosomes, micro-vesicles, ectosomes, apoptotic bodies, etc. [114]. Exosomes are a class of vesicles with diameters of 40 nm to 100 nm, which can be secreted by almost all cells and tissues in the body. The generation of exosomes is complex and diverse, and several molecules and physiological process are involved. In brief, vesicular endosomes uptake proteins or RNAs, followed by fusion with plasma membrane. Then, the vesicles (exosomes) are released from the donor cells by exocytosis [108]. Exosomes can fuse with live cells, transferring their cargo of lipids, glucose, proteins, and RNAs to the acceptor cells. To date, the functions of exosomal miRNAs have been well studied in various disease conditions, such as cancers, inflammation responses, and cardiovascular diseases. However, how RNAs (miRNAs and lncRNAs) are released into exosomes is still under investigation. Studies indicate that the specific motifs recognized by the RNA-binding proteins (RBPs) might be the determinant of miRNAs secreting into exosomes. For example, the ‘GGAG’ motif involved in miRNAs is bound by hnRNPA2B1 (a heterogeneous nuclear riboprotein), which directs miRNA trafficking to exosomes [115]. 

One database, named exoRBase, indicates that almost 15,500 lncRNAs have been identified in human blood exosomes [116]. How these lncRNAs are secreted into the exosome is rarely illuminated. Specific motifs present in certain lncRNAs may guide their sorting to exosomes through the interaction with specific RNA-binding proteins. Other reports find that lncRNA sorting to exosomes is regulated by changes of targeted transcript levels in the receiving cells [117]. Studies also indicate that lncRNAs with 3′ end uridylated appear over-represented in exosomes [118]. In addition, lncRNAs may also conversely regulate the secretion of exosomes. The biogenesis and secretion of exosomes involve various sorting machineries, including endosomal sorting complex required for transport (ESCRT)-dependent processes [119]. Phosphorylation of synaptosome associated protein 23 (SNAP23) increases exosome production and secretion [120]. For instance, lncRNA HOTAIR promotes the secretion of exosomes through inhibiting VAMP3 and SNAP23 colocalization to induce multivesicular bodies (MVBs) fusion with plasma membrane [121].

Although exosome-associated lncRNAs have been recently reported with potential roles in the pathogenesis of disease such as cardiac fibrosis, some concerns still need to be considered when studying the functions of lncRNAs resident in exosomes. Unlike miRNAs, lncRNAs always have a long sequence or complex secondary structure, which may limit their secretion into exosomes. LncRNAs usually bind with several proteins or miRNAs, which may hinder their secretion into exosomes. In addition, the present evidence about lncRNAs resident in exosomes is usually obtained from qPCR experiments, which might make the results unreliable.

### 2.6. LncRNAs Serve as Potential Therapeutic Targets

In the last few years, many efforts have been made on the application of RNA-based therapeutics in clinical practice. Chemically modified oligonucleotides and cellular RNAs are important parts of lncRNA-targeted therapeutics [122]. Antisense oligonucleotides (ASOs), a set of single-stranded DNA molecules, are complementary to target mRNA. ASOs cause mRNA degradation or pre-mRNA splicing to block protein translation [123,124]. An interesting development in biological progress is the use of natural antisense transcripts oligonucleotide. Several lncRNAs are the antisense transcripts of coding genes and are fully complementary to the target mRNA, which may be applied for natural antisense oligonucleotide. For example, over-expression of lncRNA BDNF-AS (antisense of brain-derived neurotrophic factor, BDNF) reduces BDNF expression, while silencing of this transcript increases BDNF expression and promotes neuronal differentiation [125].

MiRNA sponges are RNA molecules that contain several specific sequences which are complementary to miRNAs [126,127,128]. MiRNA sponges can be ideal tools for loss-of-function studies in science research. For instance, miR-181-sponge containing 10 repeated complementary miR-181 sequences significantly suppresses miR-181 expression in H9c2 cells and leads to decreased production of reactive oxygen species by upregulating target mRNA mt-COX1 expression [129]. In the heart, silencing of miR-34a using miRNA sponges is protective, whilst sustained inhibition of miR-34 may be deleterious due to its tumorigenicity [130]. Obese mice benefit from miR-122 antisense oligonucleotides treatment, as reflected by the decreased plasma cholesterol levels and liver steatosis improvement [131]. The miRNA-sponge characterization of lncRNAs indicates the potential possibility for clinical treatment.

Overall, lncRNAs are attractive approaches for disease treatment in clinical practice, such as heart failure, hypertension, and cardia fibrosis. To date, no lncRNA-associated drugs have been applied in clinical trials due to the limited efficacy and potential toxicity. 

## 3. Conclusions

Myocardial fibrosis is the final step in cardiac remodeling in several cardiovascular diseases, such as myocardial infraction, diabetic cardiomyopathy, atrial fibrillation, and heart failure. Excessive fibrosis in heart tissue renders the myocardium stiffer mechanically and contributes to the deterioration of both systolic and diastolic function. Various molecules and signaling pathways are involved in the formation of cardiac fibrosis. TGF-β signaling pathway and ECMs produced by the activated fibroblasts or myofibroblasts play significant roles in the pathogenesis of cardiac fibrosis [132]. Modulation of TGF-β signaling and ECM gene expression is a vital contribution of lncRNAs in this process. However, details of the mechanisms which enable lncRNAs to regulate TGF-β or ECMs production remain a mystery. LncRNAs may partly function via chromatin modification, transcription regulation, and post-transcriptional modification in myocardial fibrosis [133]. Besides the direct roles of lncRNAs in fibroblasts or myofibroblasts, they also function through indirect ways by interacting with other cell types, for example, releasing into exosomes to alter cellular behaviors. Non-coding RNAs such as miRNAs, circRNAs, and lncRNAs have been identified as present in the exosomes derived from other cell populations (myocytes, endothelial cells, etc.). Exosome-containing lncRNAs released from the donor cells are ingested by the fibroblasts, which may influence the progress of fibrosis [134,135]. Our understanding of non-coding RNAs involved in cell-to-cell interactions in cardiac fibrosis is still relatively limited. A comprehensive understanding of the function and mechanisms of lncRNAs in myocardial fibrosis holds the key for disease prevention and treatment.

Since the discovery of lncRNAs as master regulators in cardiac fibrosis, their utilization for myocardium remodeling diagnosis and clinical treatment strategies is increasingly employed. However, no ongoing clinical trials are currently underway due to the dose-effect, off-target effects, and potential toxicity. The mechanistic importance as well as diagnostic and therapeutic utility of lncRNAs in cardiac fibrosis needs to be further studied.

## Figures and Tables

**Figure 1 biology-12-00154-f001:**
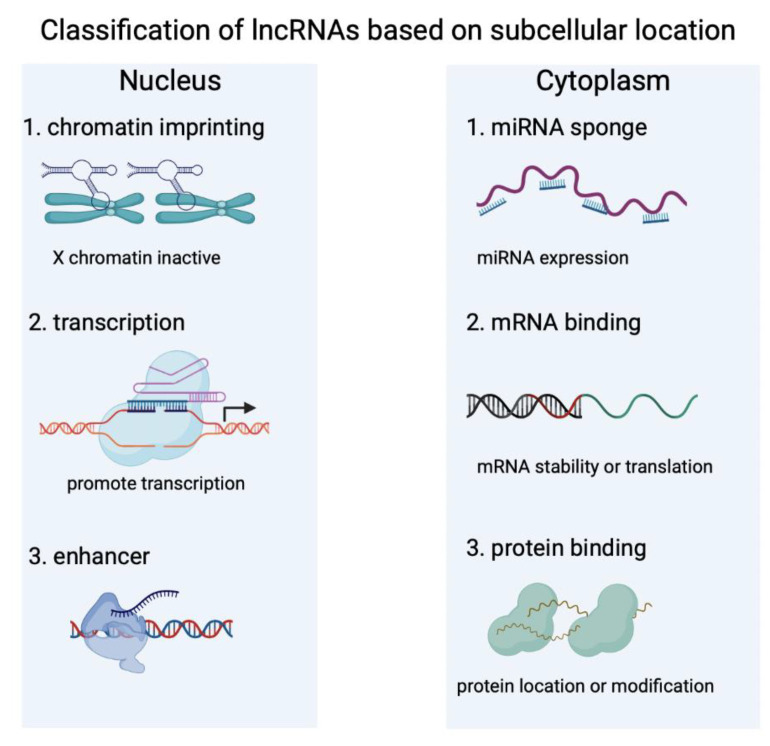
Classification of lncRNAs according to the subcellar localization. LncRNAs resident in the nucleus mainly function through regulating chromatin imprinting, controlling genes at transcriptional or post-transcriptional levels, or serving as enhancers. Cytoplasm-located lncRNAs through binding with proteins or RNAs to regulate miRNAs expression, mRNA stability or translation, and protein modification.

**Figure 2 biology-12-00154-f002:**
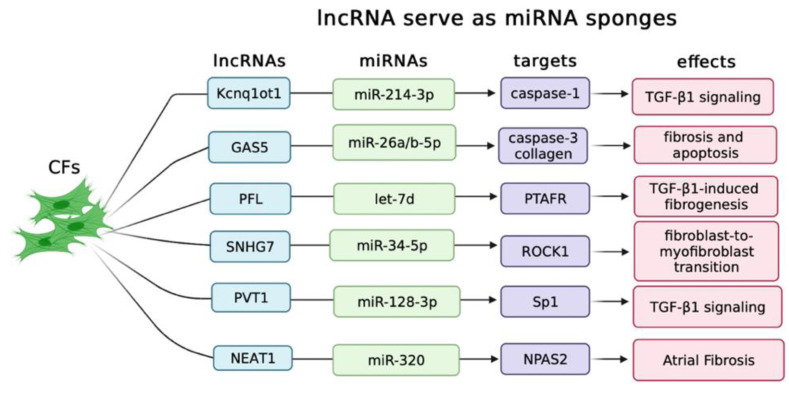
LncRNAs regulate cardiac fibrosis by serving as miRNA sponges. Various lncRNAs have been identified with crucial roles in cardiac fibroblasts by regulating miRNAs expression, which subsequently controls the target gene’s expression, and eventually regulates cardiac fibrosis.

**Figure 3 biology-12-00154-f003:**
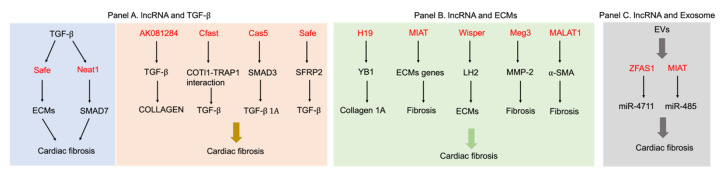
LncRNAs regulate cardiac fibrosis through promoting ECMs production, activating TGF-β signaling pathways or secreting into exosomes. Panel A: On the one hand, TGF-β regulates lncRNAs expression (left) and on the other hand lncRNAs also stimulate TGF-β activation (right). Panel B: LncRNAs regulates ECMs (collagens, SMA, and MMPs) production through various pathways. Panel C: Exosome-derived lncRNAs released from donor cells are accepted by fibroblasts and regulate cardiac fibrosis by targeting miRNAs.

**Table 1 biology-12-00154-t001:** The functions and mechanisms of lncRNAs in cardiac fibrosis.

LncRNA Name	Expression	Experimental Model	Targeted Genes	Effects	Exosomes
Kcnq1ot1	Up-regulated	DCM	miR-214-3/Caspase1	TGF-β1 signaling	No
GAS5	Up-regulated	DCM	miR-26a/b-5p/ Caspase3	Fibrosis and apoptosis	No
PFL	Up-regulated	MI	Let-7d/PTAFR	TGF-β1-induced fibrogenesis	No
SNHG7	Up-regulated	MI	miR-34-5p/ROCK1	Fibroblast-to-myofibroblast transition	No
PVT1	Up-regulated	AF	miR-128-3p/SP1	TGF-β1 signaling	No
NEAT1	Up-regulated	AF	miR-320/NPAS2	Atrial fibrosis	No
AK081284	Up-regulated	DCM	TGF-β1	Collagen I and III production	No
CFAST	Up-regulated	MI	COTL1	Enhances TGF-β signaling	No
SAFE	Up-regulated	MI	SFRP2	Fibroblast to myofibroblast transition	No
H19	Up-regulated	MI	YB-1	Reduction of collagen 1A expression	No
Wisper	Up-regulated	MI	LH2	CF behavior and survival	No
Meg3	Down-regulated		MMP-2	Diastolic performance	No
ZFAS1	Up-regulated	DCM	miR-4711-5p	Wnt4/β-catenin signal pathway	Yes
MIAT	Up-regulated	AF	miR-485-5p	Atrial remodeling	Yes
Neat1	Up-regulated	MI	CDK1	Fibroblast and cardiomyocyte survival	Yes

DCM: Diabetic cardiomyopathy. MI: Myocardial infarction. AF: Atrial fibrillation.

## Data Availability

No new data are created or analyzed in this study. Data sharing is not applicable to this article.

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
