# Peer review of "The Function and Therapeutic Potential of lncRNAs in Cardiac Fibrosis"

_biology, 2023, doi:10.3390/biology12020154_

Round 1

Reviewer 1 Report

The manuscript by Nie et al. overviews an important and relevant for translational research topic with a potential future application in the clinical practice, namely the role of lncRNAs in cardiac fibrosis. Overall, the paper is well written and detailed. In order to be consider for publication some minor comments should be addressed.

Figure 3 could be elaborate and include more details of the mechanisms of lncRNA regulation through TGFb, ECM, and exosomes. The different mechanisms could be schematically present in different panels (A, B, and C) and should include the specific genes/proteins involved in fibrosis/ECM production.

Line 290-297 should be more clearly written – it should be distinct between miRNA and lncRNA in the exosomes.

Some language proof reading should be performed and some typos to be corrected – for example line 312 – “serve” instead of “sever”.

Reviewer 2 Report

This is a very nice and timely review on the functional role of lncRNAs in cardiac fibrosis. I have only one observation. It is rather weird to have a subheading of discussion, since no results are presented. I would suggest to delete such subheading and integrate this information into a conclusion/perspective subheading.

Minor comments,

in several passages of the manuscript there are distinct letter sizes. Please modify accordingly.

Reviewer 3 Report

Synopsis

This manuscript focuses on the role of LncRNAs in cardiac fibrosis. The authors have completed a comprehensive search of current research in this area. The manuscript also investigates the roles of cardiomyocyte derived EV LncRNAs modulating fibroblasts and then discusses EV in greater detail. This is indeed an interesting and under researched field of study for LncRNAs. Overall the quality of the review is good, however, the introduction is not at all as well written as the discussion and I would like to see this adjusted. Alterations should be made to the figures so that they can be fully appreciated and a table collating the LncRNAs discussed would be helpful.

Comments

Different font size needs to be addressed throughout paper

The manuscript needs to be proof read to ensure consistency in style of writing across the manuscript. Some sections are extremely well written and others which contain mistakes (I have highlighted a few in text as examples).

There is a nice body of research contained within the manuscript referring to LncRNAs in cardiac fibrosis - I suggest that these are collated into a table which detail the LncRNA, cell type investigated, disease type if in-vivo, animal, process regulated (or MIR) etc and also authors which date.

The figures are interesting and should be included however they require more explanation and relevance to the text. I have explained more below.

Careful consideration to the choice of words when describing particular processes in the introduction:

Line 42-48 is a simplified explanation of reparative and reactive fibrosis and may be due to the paper cited being circa 1987. This whole paragraph should be expanded to explain the differences between these processes and also how they are linked.

The 2 manuscripts referenced below provides more detailed descriptions which is particularly important given that this manuscript being reviewed states that reactive fibrosis is not accompanied by myocyte death – but later states that myocyte death can occur (which is correct) and therefore this paragraph needs clarification and expansion.

1)Svenja Hinderer, Katja Schenke-Layland. Cardiac fibrosis – A short review of causes and therapeutic strategies. Advanced Drug Delivery Reviews, Volume 146, 2019,Pages 77-82,

ISSN 0169-409X

2)Biernacka A, Frangogiannis NG. Aging and Cardiac Fibrosis. Aging Dis. 2011 Apr;2(2):158-173. PMID: 21837283; PMCID: PMC3153299.

Line 49-50 refers to fibroblasts being the central cellular effectors in cardiac fibrosis – however, both myocytes and macrophages are key effectors triggering initiation of cardiac fibrotic processes. Careful re-wording of the sentence is required. Perhaps consider - fibroblasts are the main cellular effectors responsible for extracellular matrix deposition.

Lines 49-56 There is a duplication with descriptions for example repeating of ECM deposition proteins by fibroblasts. The paragraph should be restructured so the paragraph reads more concisely.

Line 76-77. The sentence requires rewording as duplication of human genome and the data from encode large scale screening has not been validated across the genome for functional relevance for all sites so perhaps amending to a sentence similar to - Mapping sites of transcription, transcription factor association, chromatin structure and histone modification by the ENCODE project suggests that up to 80% of the human genome may be involved in gene regulation.

Lines 76-79 have been taken direct from another review, as shown below, it is extremely similar and therefore should be reworded.

Review paragraph

In the human genome, up to 80% of the human genome might be involved in gene regulation, whereas only ~2% of the genome seems to code for proteins. The functional noncoding genome can be broadly divided into cis-regulatory DNA elements and non-protein coding RNAs (ncRNAs)

Published review text

Large-scale genomic surveys suggest that up to 80% of the human genome might be involved in gene regulation, whereas only 2% of the genome seems to code for proteins1. The functional noncoding genome can be broadly divided into cis-regulatory DNA elements and regions that encode non-protein coding RNAs (ncRNAs). 

Elkon, R., Agami, R. Characterization of noncoding regulatory DNA in the human genome. Nat Biotechnol 35, 732–746 (2017). https://doi.org/10.1038/nbt.3863

Line 79-80 RNAs longer than 200 nucleotides are classified as lncRNAs - has already been stated in the introduction.

Line 81-84 It is important to mention that whilst there are a higher number of lincRNAs continually being identified, most are not validated and function largely unknown.

Line 81 Based on association with annotated protein-coding genes – this part of the sentence does not make sense and therefore is not needed.

The text for figure 1 needs to be referenced as the figure does not match the text it is associated with in line 82-89 – these referenced papers are over 10 years old and therefore more up to date references for subcellular functions should be included which are in line with the functions detailed in the figure.

Line 115-116 This statement “Several lncRNAs positively or negatively regulate acute myocardial infarction and heart failure[47]” is incorrect. Several lncRNAs have been associated with these clinical conditions or found to regulate processes associated to them or expression levels correlated to disease progression. Better references are required and the sentence should be amended.

Line 154-155 “These studies widely demonstrate that lncRNAs are elevated and function as miRNA sponges in myocardial fibrosis after heart injury”. The paragraph above this sentence reads well and contains interesting lncRNA interactions with miRNAs in different disease types impacting fibrosis. However, the sentence reads as a sweeping statement that when lncRNAs are elevated they function as miRNA sponges – I believe this is simply a language barrier and would recommend changing the sentence. Suggestion below:

“The above studies have revealed specific lncRNAs which when elevated function as miRNA sponges to mediate fibrotic processes associated with varied cardiac diseases”.

Line 157 – explore change to exploring

Line 163 – failure change to fail

Line 173 – provoke change to promote or initiate

Line 186 – In the – delete the

Line 186 – this sentence indicates that all lncRNAs are upregulated in response to TGFB both in vivo and in vitro – I do not believe is correct. There is no reference and I believe the authors are referring to the research they are reviewing in the paragraph which follows. Therefore, the sentence needs to be amended to reflect that SOME lncRNAs have been shown to be upregulated in response to TGFb.

Line 191 – underline change to underlying

Line 192 - are rarely illuminated in these biological processes. Perhaps change to is not currently known.

Line 197-198 Change to More work is still required to answer these questions.

Line 202 Which types of collagen should be stated.

Figure 3. The figure it predicted modes of mechanism and should clearly state this. Also Figure 3 is referred to in the text but doesn’t match the image. Figure 3 should be moved to after discussing ECM and exosomes in the manuscript and a more detailed figure legend is required to explain the proposed mechanisms based on the reviewed manuscripts contained within the review. It is an interesting finding based on this review and should be explained more fully.

Line 238-241 The manuscript referenced here should be explained in greater detail. The authors did n=15 study comparing hearts of ICM to controls and identified 145 differentially regulated lncRNAs from which they tested the ECM genes, proteins mentioned for 5 of these LncRNA in mouse CF.

Line 262-263 Since the discovery that non-coding RNAs such as miRNAs and lncRNAs are present in human body fluids, partly via the inclusion into extracellular vesicles (such as exosomes and microvesicles)[89, 90].  Neither of the manuscripts referenced specifically mention LncRNAs in exosomes they both refer to microRNAs – reference incorrect for statement.

Line 262 – 265 “Since the discovery that non-coding RNAs such as miRNAs and lncRNAs are present in human body fluids, partly via the inclusion into extracellular vesicles (such as exosomes and microvesicles)[89, 90]. They have been emerged as paracrine effectors by which car-264 diac cell types communicate with each other and respond to stress conditions[91, 92].” The first sentence is not structurally correct and requires rewording.

Line 268-269 “Researchers report a crosstalk between cardiomyocytes and fibroblasts mediated by the transfer of lncRNA-enriched extracellular vesicles” This statement needs to be referenced

Line 270-274 The manuscript referenced here has been WITHDRAWN from publication and therefore this paragraph cannot be included in review.

Line 257 Exosome-derived lncRNAs regulate cardiac fibrosis – based on the above notes – this section needs to be revisited and amended. However, based on Kenneweg et al there is crosstalk driven via hypoxic cardiomyocyte EV and LncRNA NEAT1 on fibroblasts which is certainly interesting but needs to be explained much better, perhaps by explaining the experimental design and execution of the research and results. Since there is to date little research in this area – this is why I feel you need to be much more specific about Figure 3 and mode of mechanisms.

Line 186 Remove a closing bracket

Round 2

Reviewer 3 Report

The revised draft of this review is very good and informative. The entire reviews now flows nicely and the figures and table are particularly helpful. Original feedback has been addressed.

I would still suggest one final proof read prior to final submission as I noticed some typos (of which I have extracted and highlighted some blow)

Line 10 – lead to leads

Line 21 – should first be typed in full as extracellular matrix (ECM)

Line 43 – matrix should be change to matrix protein

Line 44 – FN should be first typed in full as fibronectin (FN)

Line 75 genom – misspelt should be demone

Line 76 The transcripts of most of these genomes (do you mean genes?

Line 78 mammaliantranscriptome – requires a space between word

Line 88 gene requires a S

Line 101/102 – gap in sentence

Line 252 various changed to variety or selection ?

Line 255 – grammar -  In brief, proteins or RNAs 255 are uptake by the vesicular endosomes, (perhaps reword to-  In brief, vesicular endosomes uptake proteins or RNAs,

Line 260 -  released into exosomes are still under exploring – requires ‘are’ at start of this

Line 318 – discovering should be discovery
